# No Man is an Island: The Impact of Neighborhood Disadvantage on Mortality

**DOI:** 10.3390/ijerph16071265

**Published:** 2019-04-09

**Authors:** Darrell J. Gaskin, Eric T. Roberts, Kitty S. Chan, Rachael McCleary, Christine Buttorff, Benjo A. Delarmente

**Affiliations:** 1Department of Health Policy and Management, Johns Hopkins Bloomberg School of Public Health, Baltimore, MD 21205, USA; rmcclear@jhu.edu (R.M.); bdelarmente@jhu.edu (B.A.D.); 2Hopkins Center for Health Disparities Solutions, Johns Hopkins Bloomberg School of Public Health, Baltimore, MD 21205, USA; Kitty.S.Chan@medstar.net; 3Department of Health Policy and Management, University of Pittsburgh Graduate School of Public Health; Pittsburgh, PA 15261, USA; eric.roberts@pitt.edu; 4MedStar-Georgetown Surgical Outcomes Research Center, MedStar Health Research Institute and Medstar Georgetown University Hospital, Washington, DC 20007, USA; 5RAND Corporation, Arlington, VA 22202, USA; buttorff@rand.org

**Keywords:** social determinants, health, neighborhood disadvantage, mortality, socioeconomic status

## Abstract

This study’s purpose is to determine if neighborhood disadvantage, air quality, economic distress, and violent crime are associated with mortality among term life insurance policyholders, after adjusting for individual demographics, health, and socioeconomic characteristics. We used a sample of approximately 38,000 term life policyholders, from a large national life insurance company, who purchased a policy from 2002 to 2010. We linked this data to area-level data on neighborhood disadvantage, economic distress, violent crime, and air pollution. The hazard of dying for policyholders increased by 9.8% (CI: 6.0–13.7%) as neighborhood disadvantage increased by one standard deviation. Area-level poverty and mortgage delinquency were important predictors of mortality, even after controlling for individual personal income and occupational status. County level pollution and violent crime rates were positively, but not statistically significantly, associated with the hazard of dying. Our study provides evidence that neighborhood disadvantage and economic stress impact individual mortality independently from individual socioeconomic characteristics. Future studies should investigate pathways by which these area-level factors influence mortality. Public policies that reduce poverty rates and address economic distress can benefit everyone’s health.

## 1. Introduction

“Your zip code may be more important to your health than your genetic code” [1]. The importance of “place” has galvanized public health researchers, advocates, and practitioners to study, promote, and implement neighborhood-based policies to improve health. Studies in public health and social science literature document the association between neighborhood factors and health [2,3,4,5]. Ecological studies have found significant differences in mortality rates and life expectancy across geographic areas [6]. Recent evidence has shown that mortality rates and life expectancy varied significantly across areas within the United States, especially for low-income individuals [7]. The authors concluded that most of the area variation in life expectancy is associated with variation in health behaviors (i.e., smoking, obesity, and exercise) and did not find support for other area level factors, such as access to medical care, pollutions, labor market conditions and economic inequality [7]. However, there is evidence of local area variation in mortality and life expectancy. Even zip codes that are in relatively close together can have huge differences in life expectancy. For example, the Baltimore City Health Departments reports a 20-year difference in life expectancy across neighborhoods within the city [8].

In theory, neighborhoods’ impact mortality through their physical and social environments [9]. The impact of the physical environment can be characterized in two ways. First, where someone lives determines their exposure to characteristics of the physical environment that impact health, such as pollution, housing, infrastructure, and the built environment. These environmental factors include air and water quality, exposure to hazardous materials, such as lead, asbestos, and industrial waste, and general safety. Air quality has been linked to cardiovascular disease and cancer. For example, a 10 microgram increase in airborne fine particulate matter is estimated to increase cardiovascular-related hospital admissions by 0.64–0.68% [10]. A recent study found that the level of small particulate matter is associated with higher county mortality rates [11]. A study of 107 U.S. cities showed that the prevalence of vacant or boarded up houses is associated with a higher risk of premature mortality due to diabetes, homicide, and suicide [12]. Second, where someone lives determines their access to community level resources that can improve health, such as the availability of healthy foods, recreational facilities, healthcare, education, and transportation. Social environment refers to social norms, social cohesion, social connections, and social stressors. Researchers have measured social environment with crime statistics, household composition, neighborhood socioeconomic and racial-ethnic composition, and resident surveys of safety, cohesion, efficacy, and disorder [9]. For example, researchers have identified relationships between resident perceptions of neighborhood safety with physical activity and medication adherence [13,14,15]. Violent crime is associated with higher county-level mortality, while the percentages of married households and Hispanic and Asian residents are predictive factors [11].

Neighborhood disadvantage is a major contributor to racial and socioeconomic disparities in mortality and health status. Because African Americans and poor persons tend to live in resource deprived neighborhoods with greater environmental, social, and economic health risks, race and socioeconomic differences observed in national mortality statistics may reflect the influence of neighborhood-level disadvantage instead of just differences in individual-level factors. The Moving to Opportunity (MTO) Study found evidence that living in a high poverty neighborhood increased the risks of diabetes and extreme obesity for low income adults and lowered their subjective well-being [16,17,18]. Additional results from the MTO Study have shown that moving to a lower poverty area improved children’s long-term economic outcome as they were more likely to attend college and had substantially higher income as adults. However, moving to a lower poverty area did not improve the economic outcomes of adults. Outside of the context of the MTO, studies have also shown that children’s long-term economic outcomes improve when they move to more prosperous neighborhoods and counties [19,20]. To our knowledge, most of the research on the neighborhood disadvantage and health discusses the adverse impact of residential and income segregation on health among low-income persons and minorities. These studies have informed the development of public policies, such as Hope VI, Choice Neighborhood, Promise Zones Initiative, and other programs implemented by the U.S. Department of Housing and Urban Development, which aim to improve the social determinants of health for disadvantaged groups. 

However, there is an outstanding question regarding whether neighborhood disadvantage negatively affects the health of more affluent persons. This article examines the impact of neighborhood disadvantage on the risk of mortality for purchasers of term life insurance policies, a more affluent sample compared to MTO subjects and the U.S. population in general. Specifically, we determine if among term life insurance policyholders life expectancy is negatively associated with neighborhood disadvantage, poor air quality, economic distress, and violent crime rates after adjusting for individual demographic, health status, and socioeconomic characteristics. This allows us to assess whether neighborhood disadvantage has a negative effect on life expectancy for individuals with relatively high socioeconomic status.

## 2. Materials and Methods

### 2.1. Data

This study uses 2013 mortality data from term life policies issued in the U.S. by a large national life insurance company from 2002 to 2010. Term life insurance policies are purchased for a fixed annual premium for a set period, such as 10 or 20 years, with benefits paid only if the policyholder dies within the specified term. From 1.69 million active policies issued from 2002–2010 we identified 10,247 death claims that occurred by 2013, for an overall mortality rate of 0.61%. The life insurance company does not maintain policyholders’ applications in an electronic database, but original paper copies and scans of applications are available. We asked the insurance company to review its records of policyholders’ applications and abstracted a comprehensive set of individual characteristics for each policyholder. To save money, we attempted to obtain comprehensive data for all deceased policyholders and a sample of live policyholders. For each deceased policyholder, we randomly selected four live policyholders who purchased their policy in the same year as the deceased policyholder. Of the 10,247 deceased policyholders, we obtained comprehensive data for 7734. Among the 40,494 randomly selected live policyholders we obtained comprehensive data for 30,338. All policies are observed for periods ranging from less than three to ten years after date of issue. All data collection procedures were approved by the Institutional Review Board at Johns Hopkins Bloomberg School of Public Health (IRB #3808 *).

We obtained the following information from the policyholders’ application: age, gender, annual personal income, and occupation. Policyholders used free text to indicate their occupation on the insurance application. We used their descriptions to assign them to 2010 U.S. Census occupational categories. We searched the occupational status fields in the database for descriptors used by the Census Bureau under each major occupational category. We were able to assign 60% of the sample to an occupation. In addition, participants in the database self-reported smoking status and health conditions: diabetes, cancer, asthma, depression, high blood pressure, previous heart attack, stroke, as well as height and weight. Family history of cancer and heart disease were also collected, including whether a family member died from either condition. The application does not contain information about the policyholders’ race or ethnicity, as life insurance companies are prohibited from collecting this information. However, our sample likely has a higher percentage of whites compared to the general population because whites (36%) purchase term life policies at a higher rate than blacks (23%) [21].

Most importantly, the application contains the policyholder’s zip code and county of residence at the time the policy was issued. We used this information to link zip code level measures of socioeconomic status and county level measures of air quality, economic distress, and violent crime. We obtained zip code level socioeconomic measures from the 2000 and 2010 U.S. Census and the 2011 five-year estimated American Community Survey. Specifically, we used the following variables: poverty rate, near poor rate (i.e., percent of the population between 100 and 199 of the federal poverty level), percentage of adults with no high school education, percentage of adults in non-professional and non-managerial occupations, unemployment rate, and percentages of renter-occupied and vacant homes. Also, we used the racial and ethnic composition of the zip code to proxy individual races and ethnicities. 

The county level information came from the Environmental Protection Agency (EPA), the Federal Bureau of Investigation’s Uniform Crime Reporting System, and the New York Federal Reserve. The EPA maintains several measures of air quality on carbon monoxide, nitrogen dioxide, ozone, sulfur dioxide, and particle matters (PM_2.5_ and PM_10_). In our analysis, we used the annual average of “peak” daily readings of small particulate matter (PM_2.5_), which measures particles less than or equal to 2.5 micrometers. We did not use the other air quality measures, because they were missing for many of the counties represented in our data. PM_2.5_ has been shown in prior studies to be related to mortality [10]. The FBI maintains rates of violent crimes per 1000 residents. Violent crimes are defined as murder, forcible rape, robbery, aggravated assault, property crime, burglary/larceny, and motor vehicle thefts. Finally, we used county rates of mortgage delinquencies over 90 days in arrears reported by the NY Federal Reserve to measure economic distress.

### 2.2. Measuring Neighborhood Disadvantage

We created a composite measure of neighborhood disadvantage to facilitate assessment of the influence of neighborhood socio-economic context on an individual’s risk of mortality [22]. We used factor analysis to develop a composite neighborhood disadvantage score comprised of the following neighborhood socio-economic status (SES) variables from the Census in 2000 and 2010: the percentage of adult residents with no high school education, the percentage unemployed, the percentage of non-white collar and non-professional jobs, the percent living in poverty, and the percentage of homes that were either vacant or renter-occupied. Prior research on the effects of place on health have similarly used factor analysis with these SES variables in the construction of a composite neighborhood disadvantage [23] or area deprivation index [24,25]. Based on eigenvalue = 1 criterion and a scree-plot analysis, we found a one-factor solution, which indicates that valid factor scores of neighborhood disadvantage could be obtained (See Figure 1). Standardized versions of the variables were created using the z-score method of subtracting the mean from each variable and dividing by its standard deviation. Average correlations of this set of variables were 0.48 and Cronbach’s alpha was 0.82. These variables were multiplied by their factor loadings to create an aggregate factor score [22]. 

### 2.3. Statistical Analysis

We estimated multivariate Cox proportional hazard models to examine the relationship between person- and area-level characteristics and mortality. We can observe the policyholders from the time they bought their policy until their death or the end of the observation period in which they are alive. Therefore, the data lends itself to survival analysis. The Cox proportional hazards model takes advantage of the time duration between the time that policyholders bought their policy to the time of their deaths or the end of the observation period. We did not use probit and logit regression models because they ignore the time element in the data. We used multiple imputation to fill-in missing area-level data, using all available area-level covariates to impute missing values [26,27]. We had missing data for measures of air quality (25%), economic distress (6%), and violent crime (6%). About 15% of policyholders were missing personal income data. We imputed income using a semi-log model with occupational status, age, gender, region, and urban-rural location as explanatory variables. This model explains 18.2% of the variation in income among the non-missing observations. 

We estimated two main hazard models. The first main model included all person-level variables, neighborhood disadvantage, fine particulate matter, mortgage delinquency, and violent crime. In the second main model, we replaced the neighborhood disadvantage score with the components of the score. Additionally, we estimated three other specifications for each of the two main hazard models. The first additional specification included only area-level variables. Next, we estimated hazard ratios with only person-level variables. Finally, we estimated a model that included both area-level and person-level variables but excluded the individual SES variables for occupation, age, and income. Estimating these models allowed us to judge the robustness of the associations. Specifically, we were interested in how the estimated hazard for neighborhood disadvantage changes when we included and exclude individual SES. 

We were concerned about unobserved factors that influence neighborhood choice and the risk of dying. Because policyholders can move, thereby changing their exposure to the area-level variable, we estimated models for all policyholders, and separately for policyholders who had moved during the time they had the policy, or before they had died. We also estimated models including the change in neighborhood disadvantage to determine if decline or improvement in neighborhood factors impacted mortality. 

We conducted both unweighted and weighted analyses. We used probability weighting for our weighted analysis to reflect the differential probabilities of selection into the sample. In particular, deceased policyholders were assigned a probability of 1 while survivors were assigned their respective probabilities of selection for their particular issue year. On average, a survivor had a 0.018 probability of selection into the sample. For sensitivity analyses, we stratified the analyses by income by dividing the sample into three groups: less than $40,000, between $40,000 and $75,000, and greater than $75,000. This stratification roughly divides our sample into thirds. 

This study has a few limitations. This is a convenient sample and the findings are not generalizable to the U.S. population. People who buy life insurance policies are more affluent and tend to have assets they are trying to preserve for their heirs in case they die unexpectantly. The analysis may be subject to selection bias. There may be unobserved factors that influence neighborhood choice and the risk of dying. We have tried to minimize this by randomly selecting surviving policyholders and including all deceased policyholders in the analysis. We also measured individual and neighborhood factors at baseline instead of contemporaneously.

## 3. Results

### 3.1. Description of the Sample

The mean duration for policyholders in the sample was 4.9 years. By design, 20% of the sample died during the study period, compared to 0.61% of policyholders in the study population. Policyholders in the sample were more affluent compared to adults nationally (See Table 1). Their mean personal income exceeded $123,000. Only 26.9% of policyholders in our sample had personal incomes below $40,000 compared to 66.3% for the adults nationally in 2013. Also, more than 37% of the policyholders in our sample had personal incomes greater than $75,000 compared to 13.5% of adults nationally. Only 41% of policyholders were female (See Table 2). Most policyholders (87%) purchased their life insurance between the ages of 20–59. Not surprisingly, policyholders were relatively healthy; 8.9% reported they had high blood pressure, high cholesterol, heart disease, heart attack chest pain or some other heart problem, 2.9% had asthma, 1.5% had diabetes, 1.5% had cancer, and less than one percent had depression. We display the mean values of the area level variables in Table 2. By construction, the neighborhood disadvantage score is centered on zero.

We compared the zip codes of policyholders to the remaining zip codes in the nation (See Table 3). Policyholders were more likely to reside in zip codes that had higher proportions of minority residents, 75% versus 85% white. Policyholders lived in zip codes that had higher levels of educational attainment, with 24.7% versus 14.4% with a college degree and 18.4% versus 22.9% with less than a high school diploma. Policyholders lived in zip codes with lower poverty rates, 11.1% versus 14.1%. We also compared movers to those who stayed in their original zip codes (See Table 4). While there are statistically significant differences due to the large samples, there does not appear to be any meaningful differences between movers and stayers, with the exception that the movers were younger than the stayers.

### 3.2. Association of Neighborhood Disadvantage and Other Variables

Our full unweighted model shows that neighborhood disadvantage increased the hazard of dying thus having a negative association on life expectancy (See Table 5). The hazard of dying for policyholders increases by 9.8% (CI: 6.0–13.7%) as neighborhood disadvantage increased by one standard deviation. This hazard increases to 14.5% (CI: 8.2–21.2%) in our weighted model. The hazard of dying also increases with mortgage delinquency rate; a ten-percentage point increase in the delinquency rate increased the hazard of dying by 20.6% (CI: 13.1–27.4%). However, this effect becomes statistically insignificant in the weighted model. County level air pollution and the violent crime rate had a positive but statistically insignificant association with the hazard of dying. Furthermore, the model shows that individual occupation status and income impacts the hazard of dying. Compared to persons in management positions, those in science, education, and health occupations had lower hazards of dying, while persons who were not working or who did not report an occupation had higher hazards of dying. The income-mortality gradient is consistent with prior studies. The hazards of dying for persons with income greater than $120,000, and between $75,000 and $120,000 were, respectively, 17.7% and 15.9% lower than for persons with income less than $28,000. This model includes controls for age, gender, presence of chronic health conditions, and zip code level, race, and ethnicity. These control variables had the expected impact on the hazard of dying. Older persons, men, persons with chronic health problems, and those living in zip codes with higher percentages of African Americans (a proxy for individual-level race) had higher hazards of dying. Similar results were observed in the weighted model (See Table 6). 

Our other unweighted specifications show a more negative impact of neighborhood disadvantage on mortality. In our model with only area-level variables, the hazard of dying for policyholders increased by 20.9% (CI: 16.9–25.1%) as neighborhood disadvantage increased by one standard deviation. On the other hand, the effect somewhat diminishes to 11.2% (CI: 7.4–15.1) when non-SES person-level variables are included. Coefficients for the variables in our model with only person-level variables were similar to those observed in the full model. The weighted models show slightly higher magnitudes but similar trends to those found for the unweighted models. These results suggest that the negative effect of neighborhood disadvantage on mortality is stable and is not fully accounted for by individual-level characteristics.

We also examined whether the impact of neighborhood disadvantage on the hazard of dying varied for persons who moved compared to persons who lived in the same zip code for the observation period (See Table 7). The hazard in the unweighted model was greater for policyholders who moved from their issue date zip code (17.1%—CI: 8.6–26.3%) compared to policyholders who stayed in their issue date zip code (7.8%—CI: 3.6–12.2%). The same holds true in the weighted model (23.4%—CI: 10.3–38.1% vs. 11.5%—CI: 4.5–18.8%). However, the confidence intervals for these estimates overlap. We explored whether the change in neighborhood disadvantage affected the hazard of dying and did not find a statistically significant effect. 

The relatively large mortality hazards for policyholders who moved were unexpected. We hypothesized that persons would move to neighborhoods that lowered their risk of dying. We examined policyholders’ propensity to move for evidence of selection bias associated with observed baseline characteristics. Policyholders who moved had higher neighborhood disadvantage scores at baseline and moved to somewhat less disadvantaged neighborhoods. At the baseline, movers were more likely to live in urban areas with a higher percentage of white, non-Hispanic residents, and were younger and more affluent than non-movers. However, baseline characteristics explained little variation in the propensity to move (pseudo R^2^ = 0.029). Unobserved factors may be associated with moving and mortality. These factors could include adverse health events that occurred after a policy was issued, which were uncorrelated with baseline characteristics but affected the probability of moving and subsequent mortality. These factors could amplify the association between neighborhood disadvantage, mortgage delinquency, and the hazard of dying. Persons who suffer a life-threatening or major health incidences might also have a high probability of moving to less affluent areas. Consequently, this choice would increase the hazard of dying associated with neighborhood disadvantage. These results were robust for our different sensitivity analyses, dividing the sample into different income groups. The association between neighborhood disadvantage and mortality were similar across the different income groups (See Table 8). 

The hazard in the unweighted model was greatest for those who had annual personal incomes above $75,000 (11.8%—CI: 4.2–19.8%) and lowest for those who had annual personal incomes less than $40,000 (6.6%—CI: 0.2–13.4%). Similar findings were seen in the weighted model for those with incomes between $40,000–$75,000, and above $75,000 (12.7%—CI: 1.9–24.6% vs. 18.1%—CI: 6.6–30.9%). The hazard of dying for those with incomes less than $28,000 was not found to be statistically significant in the weighted model. This is interesting because while individual income is protective, it does not reduce association of neighborhood disadvantage on hazard of dying for relatively affluent policyholders. Affluent policyholders experienced both a protective individual income effect and a harmful neighborhood disadvantage effect. 

## 4. Discussion

Neighborhood disadvantage increased the hazard of dying for term life insurance purchasers after controlling for individual socioeconomic and health status. Personal income and occupational status did not insulate our relatively affluent sample of policyholders from the effects of neighborhood or community level factors. This implies that the socioeconomic composition of neighborhoods and their economic stability are important determinants of health—even for affluent persons.

Neighborhood disadvantage matters for everyone because it determines the geographic availability of community amenities and exposure to community hazards. A possible explanation for this relationship is that neighborhood disadvantage represents community level economic and political power. The socioeconomic status of one’s neighbors helps to determine which goods and services are available in a community. Private interests will only enter a geographic market if there are sufficient numbers of customers who want, and who are able to buy, their products, to generate a reasonable rate of return. Community-level socioeconomic status also influences government priorities and policies. For example, the local tax base limits local governments’ abilities to provide public services. Politicians may also be more sensitive to the concerns and needs of communities that vote and can help finance elections. 

Of the factors used to create the neighborhood disadvantage score, we found that both neighborhood poverty rate and the percentage of adults employed in non-managerial positions influenced the hazard of dying. The causal link between neighborhood poverty and mortality is fairly straightforward. According to sociologists and economists, neighborhood poverty rate is an indicator of the access and quality of public services, such as education, transportation, public safety, healthcare, and sanitation. However, the causal link between the percentages of adults employed in non-managerial positions is less clear. We offer a few possible explanations that suggest that there may be positive externalities associated with neighbors who have managerial experience. There is evidence that residents living in more affluent neighborhoods have access to social networks with greater social capital [28]. Faith and community-based organizations that impact neighborhood quality may be more effective with board members and volunteers who have some managerial experience. Managers may also have better access to public and private decision makers who can influence neighborhood quality [29]. The skills and knowledge required for managerial positions may translate into effective advocacy for education, public safety, and zoning regulations.

Beyond neighborhood disadvantage, we found that mortgage delinquency matters. The causal link between the mortgage delinquency rate and the hazard of dying warrants some discussion. We offer four possible explanations. One, the mortgage delinquency rate could be a measure of local economic downturns that impact neighborhoods. Economic downturns create stress for neighborhood residents who may worry about the quality of their community declining and their ability to maintain their own households. Stress is a known risk factor chronic disease and mortality [30,31]. Two, the mortgage delinquency rate may measure the quality of housing in a neighborhood or the homeowners’ ability to maintain their homes. Studies suggest that housing quality is positively correlated with good health. Three, the mortgage delinquency rate could indicate a wealth effect. As mortgage delinquency and foreclosure rates rise, and as the values of homes and other property in a neighborhood fall, the wealth of homeowners and local businesses diminishes, which could limit individuals’ ability to finance investments in goods and services that improve health. This finding may be evidence of a health-wealth gradient. Four, the mortgage delinquency rate may be an indication of the availability of capital for a neighborhood. Banks may be reluctant to make commercial and residential loans in communities with high mortgage delinquency rates. Consequently, communities with high mortgage delinquency rates will find it difficult to attract public and private investments to support the provision of private and public amenities that support the social determinants of health, e.g., the availability of social services, recreational facilities, schools, businesses, housing, roads, and other aspects of the built environments.

We found that air pollution had a positive but statistically insignificant impact on the hazard of dying. Prior studies have found that air pollution, specifically small particle matter, increases mortality [11,32,33,34,35,36]. The causal pathway is clear. Hazardous toxins in the environment that cause disease and death have a negative biological impact on residents [10,37]. For a sensitivity analysis, we used an alternative measure of air pollution, the number of days for which PM_2.5_ exceeded 35 micrograms per cubic meter, which is the EPA’s standard for this pollutant [38]. Our results did not change. We believe that our results are statistically insignificant because we are measuring exposure at the county level, ignoring important within-county variation. Also, we imputed 25 percent of this data.

## 5. Conclusions

Our study provides evidence that neighborhood disadvantage and economic stress impact individual mortality independently from individual socioeconomic characteristics. This is not a phenomenon restricted to poor people. Our findings suggest that affluent adults are also affected by community-level economic factors. Future studies should investigate pathways by which these neighborhood level factors influence mortality. In particular, researchers should explore the biological pathways by which a seemingly “private” event, like a mortgage default, affects mortality at the community level, and how these economic shocks affect leading causes of death, including heart disease, cancer, respiratory disease, and accidents. From a policy perspective, our findings indicate people should be concerned about the economic wellbeing of their neighbors, not only because it is the “right thing to do,” but also because it affects their own health. As the English poet John Donne wrote, ‘No man is an island entire of itself…. And therefore never send to know for whom the bell tolls; It tolls for thee.’ Public policies and private investments that lower poverty rates neighbors and ourselves. 

## Figures and Tables

**Figure 1 ijerph-16-01265-f001:**
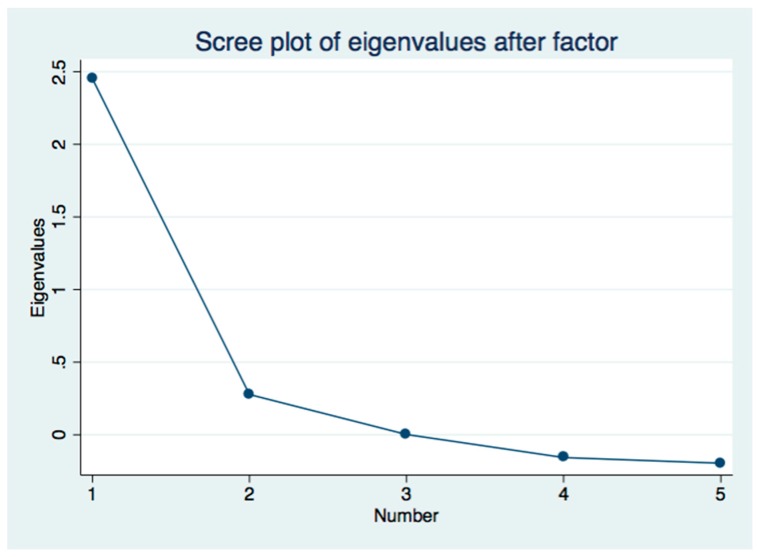
Scree Plot for Neighborhood Advantage Score.

**Table 1 ijerph-16-01265-t001:** Distribution of policy holders by income and occupational status.

Individual Level Measures	Percent	Cumulative Percent
Income < 28,000	14.20	14.20
$28,000 to 40,000	12.74	26.94
$40,000 to 54,000	18.16	45.10
$54,000 to 75,000	17.15	62.26
$75,000 to 120,000	18.98	81.24
Greater than 120,000	18.76	100.00
**Occupational Status**		
Manager	15.75	15.75
Financial	4.19	19.94
Science	3.64	23.58
Law/Social	2.59	26.17
Education	4.57	30.74
Protective Services	1.24	31.98
Health	5.77	37.75
Entertainment	1.10	38.84
Sales	5.85	44.69
Labor	9.94	54.64
Service	4.10	58.74
Not working	1.24	59.98
Other	40.02	100.00

Source: Authors’ calculations based on data set from large life insurer.

**Table 2 ijerph-16-01265-t002:** Means and standard deviations of individual and area-level variables.

Individual Characteristics	Mean	Standard Deviation
Female	0.414	0.493
Age 20 below	0.016	0.126
Age 20–39	0.379	0.485
Age 40–59	0.503	0.500
Age 60–79	0.093	0.290
Age 80 and over	0.009	0.093
Any family history	0.284	0.451
Any heart problem	0.089	0.285
Diabetes	0.015	0.120
Overweight	0.091	0.288
Depression	0.001	0.022
Asthma	0.029	0.168
Cancer	0.015	0.123
Urban	0.833	0.287
Northeast	0.168	0.374
West	0.266	0.442
Midwest	0.181	0.385
South	0.384	0.486
**Area Level Measures**		
Neighborhood Disadvantage Score	0.005	0.926
Less than High School	15.0	0.107
Unemployment Rate	5.3	0.030
Non-Managerial Employment	62.2	0.131
Poor	10.8	0.080
Near Poor	15.2	0.076
Vacancy Rate	39.3	0.187
Pollution	41.388	12.140
Mortgage Delinquency	4.098	5.103
Crime	38.128	17.235

Source: Authors’ calculations based on data set from large life insurer.

**Table 3 ijerph-16-01265-t003:** Comparison between policyholders’ zip codes with other zip codes.

Number of Zip Codes	11241	20742	*p* > |*t*|
Percent White	75.0	85.0	<0.001
Percent Black	10.2	6.1	<0.001
Percent Asian	3.2	0.7	<0.001
Percent Hispanics	9.8	5.0	<0.001
Percent No High School	7.0	9.1	<0.001
Percent Some High School	11.4	13.8	<0.001
Percent High School	29.2	37.4	<0.001
Percent Some College	27.5	25.2	<0.001
Percent College	24.7	14.4	<0.001
Poverty Rate	11.1	14.1	<0.001

Source: Authors’ calculations based on data set from large life insurer.

**Table 4 ijerph-16-01265-t004:** Comparison between policyholders who stayed and moved away.

Type of Policyholder	Stayers	Movers	*p* > |*t*|
Proportion	78.24%	21.76%	
Individual Characteristics			
Female	0.418	0.402	0.010
Age 20 below	0.015	0.019	0.003
Age 20–39	0.356	0.446	<0.001
Age 40–59	0.524	0.452	<0.001
Age 60–79	0.096	0.076	<0.001
Age 80 and over	0.008	0.006	0.018
Any family history	0.294	0.250	<0.001
Any heart problem	0.094	0.073	<0.001
Diabetes	0.015	0.013	0.188
Overweight	0.091	0.089	0.729
Depression	0.001	0.001	0.358
Asthma	0.029	0.029	0.983
Cancer	0.015	0.016	0.645
Urban	0.824	0.860	<0.001
Northeast	0.171	0.156	0.002
West	0.264	0.279	0.006
Midwest	0.185	0.172	0.009
South	0.381	0.393	0.044
**Area Level Measures**			
Neighborhood Disadvantage Score	0.005	0.003	0.852
Less than High School	15.0	14.9	0.265
Unemployment Rate	5.3	5.3	0.971
Non-Managerial Employment	62.6	61.3	<0.001
Poor	10.8	10.8	0.865
Near Poor	14.9	14.9	<0.001
Vacancy Rate	38.6	41.6	<0.001
Pollution	41.134	42.477	<0.001
Mortgage Delinquency	4.303	3.448	<0.001
Crime	37.386	40.550	<0.001

Source: Authors’ calculations based on data set from large life insurer.

**Table 5 ijerph-16-01265-t005:** Unweighted estimated hazards of dying by area-level factors, individual occupational status, and personal income, all policyholders.

Model Specification	Full Model (1)			Area Variables No Individual Variable (2)			Individual Variables Only (3)			Area Variables No Individual SES Variables (4)		
	Hazard Ratio	95% Confidence Interval		Hazard Ratio	95% Confidence Interval		Hazard Ratio	95% Confidence Interval		Hazard Ratio	95% Confidence Interval	
Neighborhood Disadvantage	1.098 ***	1.060	1.137	1.209 ***	1.169	1.251				1.112 ***	1.074	1.151
Days PM_2.5_ over 40 Micrograms per cubic meter	1.010	0.919	1.110	1.092	0.992	1.201				1.009	0.917	1.110
Mortgage Delinquency Rate	1.202 ***	1.134	1.275	1.108 ***	1.045	1.176				1.193 ***	1.126	1.265
Violent Crime Rate	1.017	0.990	1.044	1.037 **	1.011	1.065				1.018	0.992	1.046
Occupational Status (Manager is the reference)												
Financial	0.955	0.832	1.096				0.953	0.830	1.095			
Science	0.767 **	0.654	0.899				0.756 ***	0.645	0.886			
Law/Social	0.889	0.756	1.046				0.892	0.758	1.049			
Education	0.815 **	0.704	0.943				0.817	0.706	0.946			
Protective Services	0.908	0.723	1.140				0.920	0.733	1.156			
Health	0.799 **	0.698	0.916				0.806	0.703	0.923			
Entertainment	1.070	0.837	1.368				1.071	0.838	1.370			
Sales	1.026	0.914	1.151				1.022	0.911	1.147			
Labor	1.013	0.918	1.118				1.018	0.923	1.123			
Service	1.097	0.954	1.262				1.099	0.955	1.263			
Not working	1.185 *	1.023	1.373				1.183 *	1.022	1.370			
Other	1.074	1.000	1.154				1.076 *	1.002	1.156			
Income (less than $28,000 is the reference)												
$28,000 to 40,000	0.977	0.895	1.066				0.977	0.895	1.066			
$40,000 to 54,000	0.948	0.871	1.031				0.943	0.867	1.026			
$54,000 to 75,000	0.929	0.850	1.015				0.921	0.844	1.006			
$75,000 to 120,000	0.841 ***	0.770	0.920				0.833 ***	0.763	0.910			
Greater than 120,000	0.823 ***	0.750	0.904				0.813 ***	0.741	0.893			
Income Imputed Indicator	1.076	0.996	1.163				1.064	0.985	1.150			
Female	0.677 ***	0.640	0.715				0.677 ***	0.640	0.715	0.719 ***	0.684	0.756
Age (20–39 is the reference)												
Age <20	1.267	0.698	2.301				1.289	0.710	2.341	1.327	0.732	2.408
Age 40–59	3.033 ***	2.824	3.257				3.024 ***	2.816	3.248	3.005 ***	2.800	3.226
Age 60–79	7.569 ***	6.970	8.220				7.565 ***	6.987	8.214	7.744 ***	7.140	8.398
Age 80+	14.34 ***	12.410	16.550				14.55 ***	12.61	16.79	14.51 ***	12.66	16.63
Any heart condition	1.227 ***	1.147	1.313				1.229 ***	1.149	1.315	1.245 ***	1.163	1.332
Any cancer	1.254 ***	1.098	1.433				1.257 ***	1.1000	1.436	1.257 ***	1.100	1.435
Any diabetes	1.265 **	1.091	1.468				1.274	1.098	1.478	1.261 **	1.087	1.463
% Black	1.030 ***	1.013	1.048	1.021 *	1.004	1.038	1.062	1.08	1.076	1.032 ***	1.015	1.049
% Hispanic	0.996	0.977	1.015	0.977 *	0.959	0.996	1.036 ***	1.20	1.052	0.999	0.980	1.018
% Asian	0.974	0.945	1.004	0.940 ***	0.912	0.970	0.972	0.943	1.002	0.972	0.943	1.002
% Native American/Other	1.055	0.989	1.125	1.067	1.000	1.138	1.081 *	1.015	1.151	1.056	0.990	1.126

The regression includes age, gender, presence of heath conditions, smoking, obesity, family history of health status, zip code race and ethnic composition, urban-rural location, and census region. The model excludes observations above the 99th percentile of the zip code-level proportion of individuals without a high school degree. * *p* < 0.05, ** *p* < 0.01, *** *p* < 0.001. SES, socio-economic status.

**Table 6 ijerph-16-01265-t006:** Weighted estimated hazards of dying by area-level factors, individual occupational status, and personal income, all policyholders.

Model Specification	Full Model(1)			Area VariablesNo Individual Variables(2)			Individual Variables Only(3)			Area VariablesNo Individual SES Variables(4)		
	Hazard Ratio	95% Confidence Interval		Hazard Ratio	95% Confidence Interval		Hazard Ratio	95% Confidence Interval		Hazard Ratio	95% Confidence Interval	
Neighborhood Disadvantage	1.145 ***	1.082	1.212	1.251 ***	1.200	1.303				1.164 ***	1.100	1.232
Days PM_2.5_ over 40 micrograms per cubic meter	1.078	0.925	1.256	1.140 *	1.019	1.275				1.079	0.926	1.257
Mortgage Delinquency Rate	0.972	0.891	1.060	0.989	0.921	1.062				0.958	0.878	1.044
Violent Crime Rate	1.078 ***	1.034	1.125	1.054 ***	1.021	1.087				1.080 ***	1.035	1.126
Occupational Status (Manager is the reference)												
Financial	0.992	0.829	1.188				0.977	0.818	1.168			
Science	0.750 **	0.617	0.911				0.738 **	0.608	0.897			
Law/Social	0.753	0.567	1.001				0.760	0.568	1.015			
Education	0.791 *	0.645	0.969				0.797 *	0.652	0.975			
Protective Services	0.812	0.592	1.114				0.814	0.597	1.109			
Health	0.769 **	0.648	0.913				0.767 **	0.646	0.911			
Entertainment	1.082	0.772	1.514				1.088	0.782	1.515			
Sales	1.076	0.923	1.254				1.062	0.912	1.237			
Labor	0.975	0.849	1.119				0.972	0.847	1.115			
Service	1.181	0.982	1.421				1.169	0.972	1.405			
Not working	1.039	0.747	1.443				1.037	0.749	1.437			
Other	1.123	1.016	1.242				1.120	1.014	1.237			
Income (less than $28,000 is the reference)												
$28,000 to 40,000	0.959	0.831	1.106				0.950	0.824	1.095			
$40,000 to 54,000	0.877	0.763	1.009				0.855	0.743	0.984			
$54,000 to 75,000	0.857	0.745	0.986				0.826	0.719	0.949			
$75,000 to 120,000	0.744	0.645	0.858				0.714	0.620	0.822			
Greater than 120,000	0.754	0.642	0.885				0.718	0.613	0.841			
Income Imputed Indicator	1.284	1.119	1.472				1.297	1.132	1.485			
Female	0.587 ***	0.535	0.644				0.583 ***	0.531	0.639	0.656 ***	0.606	0.710
Age (20–39 is the reference)												
Age <20	1.033	0.516	2.070				1.020	0.509	2.044	1.170	0.588	2.329
Age 40–59	3.624 ***	3.341	3.930				3.621 ***	3.340	3.925	3.546 ***	3.274	3.840
Age 60–79	15.27 ***	13.55	17.21				15.28 ***	13.56	17.21	15.30 ***	13.62	17.18
Age 80+	107.1 ***	67.48	170.0				110.8 ***	70.06	175.3	108.2 ***	69.39	168.8
Any heart condition	1.347 ***	1.198	1.514				1.346 ***	1.197	1.512	1.374 ***	1.222	1.545
Any cancer	1.508 **	1.148	1.980				1.475 **	1.118	1.946	1.481 **	1.102	1.990
Any diabetes	1.394 **	1.093	1.778				1.393 **	1.095	1.0772	1.396 **	1.097	1.077
% Black	1.035 *	1.005	1.065	1.026 *	1.004	1.048	1.084 ***	1.059	1.109	1.036 *	1.006	1.067
% Hispanic	0.998	0.969	1.027	0.973 *	0.951	0.996	1.039 **	1.016	1.064	1.003	0.975	1.033
% Asian	0.963	0.923	1.004	0.929 ***	0.896	0.964	0.950	0.909	0.992	0.959	0.919	1.002
% Native American/Other	1.103	0.968	1.256	1.130 **	1.033	1.236	1.160 *	1.019	1.319	1.108	0.978	1.255

The regression includes age, gender, presence of heath conditions, smoking, obesity, family history of health status, zip code race and ethnic composition, urban-rural location, and census region. The model excludes observations above the 99th percentile of the zip code-level proportion of individuals without a high school degree. * *p* < 0.05, ** *p* < 0.01, *** *p* < 0.001.

**Table 7 ijerph-16-01265-t007:** Estimated hazard of dying for neighborhood disadvantage for all policyholders, policyholders who stayed in their zip code, policyholders who moved to another zip code.

Model Stratification (Unweighted and Weighted)	Hazard. Ratio	*p* >|*t*|	95% Confidence Interval	
A. Unweighted				
All Policyholders	1.098 ***	<0.001	1.060	1.137
Stayers	1.078 ***	<0.001	1.036	1.122
Movers	1.171 ***	<0.001	1.086	1.262
B. Weighted				
All Policyholders	1.145 ***	<0.001	1.082	1.212
Stayers	1.115 ***	0.001	1.045	1.188
Movers	1.234 ***	<0.001	1.103	1.381

The regression models include other neighborhood factors, occupational status, personal income, age, gender, presence of heath conditions, smoking, obesity, family history of health status, zip code race and ethnic composition, urban-rural location, and census region. The model excludes observations above the 99th percentile of the zip code-level proportion of individuals without a high school degree. * *p* < 0.05, ** *p* < 0.01, *** *p* < 0.001.

**Table 8 ijerph-16-01265-t008:** Estimated hazard of dying for neighborhood disadvantage for policyholders stratified by personal income.

Models Stratified by Income (Weighted and Unweighted)	Hazard. Ratio	*p* >|*t*|	95% Confidence Interval	
A. Unweighted				
<$40,000	1.066 *	0.044	1.002	1.134
$40,000–$75,000	1.113 **	0.003	1.037	1.194
>$75,000	1.118 **	0.002	1.042	1.198
B. Weighted				
<$40,000	1.096	0.077	0.990	1.213
$40,000–$75,000	1.127 *	0.020	1.019	1.246
>$75,000	1.181 **	0.001	1.066	1.309

The regression models include other neighborhood factors, occupational status, personal income, age, gender, presence of heath conditions, smoking, obesity, family history of health status, zip code race and ethnic composition, urban-rural location, and census region. The model excludes observations above the 99th percentile of the zip code-level proportion of individuals without a high school degree. * *p* < 0.05, ** *p* < 0.01, *** *p* < 0.001.

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
