# Peer review of "No Man is an Island: The Impact of Neighborhood Disadvantage on Mortality"

_ijerph, 2019, doi:10.3390/ijerph16071265_

Round 1

Reviewer 1 Report

Using a sample of approximately 38,000 term life policyholders from a large national life insurance company, the authors examine the impact of neighborhood disadvantage on mortality of more affluent populations. The results of this study are interesting and have important policy implications on promoting public policies to improve the health of the overall population. Please find my detailed comments on each section of the paper as follows.

Line 14, page 1:  I could not find the author(s) with the affiliation #5: RAND Corporation

Line 41, page 1: There are double citations at the end of the sentence “[6] Murray 2006”.

Line 76, page 2: It should be “additional results from the MTO Study have shown that…”, not “has shown that…”

Line 76-77, page 2: Please keep the verb tense in the sentence “moving to a lower-property areas improved children’s long term economic outcomes as they are more likely to attend college and have substantially higher income.”

Line 96, page 3: If possible, please be specific about the large national insurer that the authors implied here.

Line 103, page 3: Please explain more about “a four-to-one match within the year of issue”.

I don’t understand what the authors meant by “randomly sampled policyholders who remained alive during their policy term in a four-to-one match within the year of issue”.

Line 130, page 3: Please put EPA after “Environmental Protection Agency” so that the readers would know what EPA is in Line 132.

Line 145, page 3: Please spell out what SES stands for in the first time the authors mention it in the text.

Line 156-157, page 4: Please explain Figure 1 and the message that the authors would like to deliver through the figure. I don’t know how I should read the figure.

Line 183, page 5: Please explain why the authors chose the cut-off of $75,000 for their sensitivity analyses.

Line 197, page 5: It should be “…table 2 by construction. The neighborhood disadvantage score is centered on zero.”

Line 206, page 6: Table 3, comparison between policyholders’ zip codes with other zip codes.  I wonder whether the differences in demographic characteristics between two zip codes are significant.

Line 223-224, page 7: The results “older persons, men, persons with chronic health problems, and those living in zip codes with higher percentages of African Americans had higher hazards of dying” are not shown in the Tables.

Second paragraph, page 11: I would like to know the % of persons moved compared to the % of persons who lived in the same zip code for the observation period.

First paragraph, page 12: Please be specific about “our different sensitivity analyses” that the authors conducted in this context.

Other comments:

I am not sure whether the estimates of the authors can be interpreted as the impact of neighborhood disadvantage on mortality since there are unobservable factors that may affect mortality but were not captured in the models. For example, policyholders may be risk-averse. Thus, besides purchasing the policy, they may be less likely to engage in risk-taking activities (i.e. living in a nice neighborhood, and etc.).

I also would like to see whether any significant differences in demographic characteristics between policyholders and non-policyholders in the same zip codes to determine that whether policyholders are representative of the population in the policyholders’

Author Response

Comment 1: Line 14, page 1:  I could not find the author(s) with the affiliation #5: RAND Corporation

Response 1: We have corrected Dr. Buttorff affliliation.  We also deleted misplaced email addresses for coauthors.

Comment 2:Line 41, page 1: There are double citations at the end of the sentence “[6] Murray 2006”.

Response:  We have corrected.

Comment 3:Line 76, page 2: It should be “additional results from the MTO Study have shown that…”, not “has shown that…”

Response 3:  We have corrected.

Comment 4:  Line 76-77, page 2: Please keep the verb tense in the sentence “moving to a lower-property areas improved children’s long term economic outcomes as they are more likely to attend college and have substantially higher income.”

Response 4:We have corrected.

Comment 5:Line 96, page 3: If possible, please be specific about the large national insurer that the authors implied here.

Response 5:According to our DUA we are only able to identify the insurers as a large national life insurance company.

Comment 6: Line 103, page 3: Please explain more about “a four-to-one match within the year of issue”.

I don’t understand what the authors meant by “randomly sampled policyholders who remained alive during their policy term in a four-to-one match within the year of issue”.

Response 6: We have added the following clarifying language to the paragraph.

The  life insurance company does not maintain policyholders’ applications in an electronic database; however original paper copies and scans of applications are available. We asked the insurance company to review its records of policyholders’ applications and abstract a comprehensive set of individual characteristics for each policyholder. To save money, we attempted to obtained comprehensive data for all deceased policyholders and a sample of live policyholders.  For each deceased policyholder, we randomly selected four live policyholders who purchased their policy in the same year as the deceased policyholder.  Of the 10,247 deceased policyholders, we were able to obtain comprehensive data for 7,734 policyholders.  Among the 40,494 live policyholders were were able to obtain comprehensive data for 30,338 survivors.  

Comment 7:Line 130, page 3: Please put EPA after “Environmental Protection Agency” so that the readers would know what EPA is in Line 132.

Response 7: We made the correction.

Comment 8:  Line 145, page 3: Please spell out what SES stands for in the first time the authors mention it in the text.

Response 8: We made the correction.

Comment 9:Line 156-157, page 4: Please explain Figure 1 and the message that the authors would like to deliver through the figure. I don’t know how I should read the figure.

Response 9:There are 2 key rules/things to check in factor analysis – the eigenvalues and where the elbow or bend is in the scree plot. In the scree plot, the reader should be looking for when the eigenvalue goes below 1 and where the bend in the curve lies. Both of these equal one in our preliminary analysis. This therefore suggests that there is one underlying factor that explains the variables that we have. This underlying factor is our neighborhood disadvantage.  

Comment 10:Line 183, page 5: Please explain why the authors chose the cut-off of $75,000 for their sensitivity analyses.

Response 10:Actually, we stratified the data into thirds by income.  We divided the sample into roughly three relatively equally-sized groups based on cutoffs of $40,000 and $75,000. We report the results for the this analysis in Table 7.

Comment 11: Line 197, page 5: It should be “…table 2 by construction. The neighborhood disadvantage score is centered on zero.”

Response 11: We made the correction

Comment 12: Line 206, page 6: Table 3, comparison between policyholders’ zip codes with other zip codes.  I wonder whether the differences in demographic characteristics between two zip codes are significant.

Response 12:We added this data to Table 3.

Comment 13:Line 223-224, page 7: The results “older persons, men, persons with chronic health problems, and those living in zip codes with higher percentages of African Americans had higher hazards of dying” are not shown in the Tables.

Response 13:  We added these results to Tables 5A and 5B 

Comment 14: Second paragraph, page 11: I would like to know the % of persons moved compared to the % of persons who lived in the same zip code for the observation period.

Response 14:  We added Table 4 to display this information.

Comment 15:First paragraph, page 12: Please be specific about “our different sensitivity analyses” that the authors conducted in this context.

Response 15:  We conducted two sensitivity analyses:  1) stratification by income and 2) stratification by movers and stayers.

Comment 16:I am not sure whether the estimates of the authors can be interpreted as the impact of neighborhood disadvantage on mortality since there are unobservable factors that may affect mortality but were not captured in the models. For example, policyholders may be risk-averse. Thus, besides purchasing the policy, they may be less likely to engage in risk-taking activities (i.e. living in a nice neighborhood, and etc.).

Response 16:  All individuals in our sample are policyholders. Our main result is that among policyholders, neighborhood disadvantage matters. While we agree that policyholders might be risk averse, we are not talking about policyholders VS non-policyholders. We’re just saying that neighborhood mortality is associated with mortality among policyholders. Also, neighborhood disadvantage might capture other activities that are unobserved. However, these might be correlated with neighborhood disadvantage.

Comment 17:I also would like to see whether any significant differences in demographic characteristics between policyholders and non-policyholders in the same zip codes to determine that whether policyholders are representative of the population in the policyholders’

Response 17:We don’t have information about health conditions of persons at the zip code level but we can get information on the occupational status as well as the income distribution of zip codes from Census data.  The key question is whether the persons in the sample are wealthier than the average person in their respective zip code level.  We could check median personal income, age distribution, and occupational status from Census data.

Reviewer 2 Report

It is a very well written paper with interesting findings and possible public policy implications. I really enjoyed reading this paper. However, there are a number of issues that I want to ask  prior to recommending for publication.

1- Create a descriptive table of variables and make sure to show the source of datasets in the table.

2- Why did you pick Cox- Proportional hazard model (Duration Analysis). Could you justify use of this model? Why not other models? Logistic  or Probit Model? Clearly indicate your justifications in the main text.

3- How do you address the problem of potential endogeneity in the model? How do you know if the estimated coefficients are robust? Make sure to do robustness checks with your estimates.

4- Please clearly indicate the mathematical notations of your model ( Eq. 1, 2, and etc.).

5- Could you please plot the estimated coefficients instead of tables? Place the tables in the Appendix.

6- How do you know if an estimated coefficient is statistically significant in the tables? Is it possible to show the significance levels with stars?

Author Response

Comment 1:  Create a descriptive table of variables and make sure to show the source of datasets in the table.

Response 1:  We added the sources to the tables.

Comment 2:  Why did you pick Cox- Proportional hazard model (Duration Analysis). Could you justify use of this model? Why not other models? Logistic  or Probit Model? Clearly indicate your justifications in the main text.

Response 2: We added this text to the manuscript to justify using a Cox proportional hazard model.

We can observed the policyholders from the time they bought their policy until their death or the end of the observation period in they are alive. Therefore, the data lends itself to survival analysis. The Cox proportional hazards model takes advantage of the time duration between the time policyholders bought their policy to the time of their death. The use of other limited dependent variable models such as probit and logit would ignore the time element in the data.

Comment 3:  How do you address the problem of potential endogeneity in the model? How do you know if the estimated coefficients are robust? Make sure to do robustness checks with your estimates.

Response 3:  We recognize that there may be unobservable factors that may be correlated with individual and neighborhood characterististic and the risk of dying.  However, we are estimating how baseline characteristics predict the likelihood of death in the future. The neighborhood conditions under which policyholders choose to live are made ex ante. Our analysis minimizes the endogeneity problem because it is a survival analysis as oppose to a cross sectional analysis.  We don’t expect selection bias to impact our estimates because we abstracted data for all of the decease policyholders and randomly selected survivors.  Policyholders are only exclude if the life insurance company could not find the policyholders’ original application.  Are sampling frame for the survivors insured that application year did not impact the policyholders likelihood of being in the sample.  Finally, we estimate the hazard of movers compared to stayers. We are able to determine whether the effect of location influences the likelihood of death. Given that our movers are more likely to die than our stayers (we were expecting the association to go the other way), our results suggest to us that the main results are robust.

Comment 4:Please clearly indicate the mathematical notations of your model ( Eq. 1, 2, and etc.).

Response 4:We don’t have any equations in the paper

Comments 5:Could you please plot the estimated coefficients instead of tables? Place the tables in the Appendix.

Response 5:We prefer to present the hazard ratios in a table allowing the reader to compare the models, (i.e., full, individual only, area only, area without individual SES) and see the consistency of the results across specifications.

Comment 6:How do you know if an estimated coefficient is statistically significant in the tables? Is it possible to show the significance levels with stars?

Response 6:  We present the 95% Confidence Intervals and use stars to show significance levels.

Reviewer 3 Report

I have carefully reviewed the article submitted. I do have a few major concerns regarding the study: 1. Were the models tested for multicollinearity? It does not seem apparent from the manuscript, and several predictors are potentially highly correlated (e.g., neighborhood disadvantage and mortgage delinquency) which could result in artificially inflated or biased estimates. 2. The subanalysis of persons who have moved is missing a crucial information, i.e., the timing of the move. Since we are measuring exposure to social disadvantage, it is important to have the duration of exposure. Absent more detail those results are difficult to interpret and should be noted as a major limitation. 3. Main concern: Since the goal of the study was to evaluate the effects of neighborhood disadvantage on more affluent individuals, and the sample actually includes individuals with incomes ranging from less than $28000 to over $120000, a sub-analysis of individuals that are more affluent, or a model using interactions between income and disadvantage, is necessary. The authors allude to that in the Methods section but do not present any results. In the absence of that, the only finding from the models is that both income and social disadvantage influence mortality, and the conclusion of the study is an overreach.

Author Response

Comment 1:Were the models tested for multicollinearity?  It does not seem apparent from them manuscript and several predictors are potentially highly correlated (e.g. neighborhood disadvantage and mortgage delinquency) which could result in artificially inflated or biased estimates.

Response 1: We are concern about multicollinearity between individual SES and neighborhood disadvantage.  This is why we estimate the models with and without the individual SES and neighborhood disadvantage variables.  We conducted VIF tests on the predictors that are potentially highly correlated (e.g., neighborhood disadvantage and the area variables) and they do not suggest multicollinearity problems. 

Comment 2:  The subanalysis of persons who have moved is missing a crucial information, i.e., the timing of the move.  Since we are measuring exposure to social disadvantage, it is important to have the duration of exposure.  Absent more detail those results are difficult to interpret and should be noted as a major limitation.

Response 2:  Unfortunately, the life insurance company did not keep records of when policyholders moved.  We only know if their current address is different from the address on their original application.  We note this in the limitation section of the paper.

Comment 3:Main concern:  Since the goal of the study was to evaluate the effects of neighborhood disadvantage on more affluent individual, and the sample actually includes individuas with income ranging from less than 28000 to over $120000, a subanalysis of individuals that are more affluent or a model with interactions between disadvantage is necessary.  The authors allude to that in the Methods section but do not present any results.  In the absence of that the only findings form the models is htat both income and social disadvantage influence mortality and the conclusion of the study is an overreach.

Response 3: We stratified the sample into three income groups and find that the associated hazards are greatest for the highest income group. We find that the hazard of dying was higher in the two higher income groups.  Results are reported as table 7  We added the following text to the manuscript.

The hazard in the unweighted model was greatest for those who had annual personal incomes above $75,000 (11.8% - CI: 4.2%-19.8%) and lowest for those who had annual personal incomes less than $40,000 (6.6% - CI: 0.2%-13.4%). Similar findings were seen in the weighted model for those with incomes between $40,000-$75,000, and above $75,000 (12.7% - CI: 1.9%-24.6% VS 18.1% - CI: 6.6%-30.9%). The hazard of dying for those with incomes less than $28,000 was not found to be statistically significant in the weighted model.

Round 2

Reviewer 1 Report

Thank you for revising the manuscripts and providing the responses to my previous comments. The manuscript has been improvement a lot. After reviewing the manuscript the second time, I have a few more minor comments on the revised manuscript. Please find my comments as below.

Page 2, line 47: It should be “neighborhoods impact”, not “impacts”

Page 2, line 88: It is a little bit confusing to me to understand the sentence “mortality among term life insurance policyholders is negatively associated with neighborhood disadvantage”. As I understand, the relationship between these two is that when neighborhood disadvantage increases, mortality increases. In other words, they are expected to move in the same direction. Perhaps, it would be easier to understand if the authors state that “neighborhood disadvantage is negatively associated with life expectancy”.

Page 2, line 91: It should be “negative effects”, not “a negative effects”.

Page 3, line 105: It should be “obtain”, not “obtained”.

Page 4, line 168: Perhaps, it should be “….or the end of the observation period in which they are alive”.

Page 5, line 191: It should be “…time they had the policy or had died”.

Page 5, line 219: It should be “…display the mean values of the area level variables in table 2…”.

Page 5, line 229: It should be “….younger than stayers…”

Page 8, line 240: Similar to my previous comments on the negative impact of neighborhood disadvantage on mortality.

Page 14, Table 7: In Table, the authors show that in both unweighted and weighted models, the hazard of dying is largest for those who make >$75,000 annually. However, in Tables 5A and 5B, the results suggest that the higher the income a person is, the lower the hazard of dying for that person is. This is consistent with the results suggested in Table 7.

Could the authors provide some explanations to reconcile this?

Page 16: The authors should have a separate section for conflicts of interest rather than having it in the same section with acknowledgement.

Author Response

Thank you for revising the manuscripts and providing the responses to my previous comments. The manuscript has been improvement a lot. After reviewing the manuscript the second time, I have a few more minor comments on the revised manuscript. Please find my comments as below.

Reply: Thank you very much for your valuable comments on our manuscript, we have revised our paper according to your suggestion. The following parts are our response.

Page 2, line 47: It should be “neighborhoods impact”, not “impacts”

Reply: We revised it accordingly.

Page 2, line 88: It is a little bit confusing to me to understand the sentence “mortality among term life insurance policyholders is negatively associated with neighborhood disadvantage”. As I understand, the relationship between these two is that when neighborhood disadvantage increases, mortality increases. In other words, they are expected to move in the same direction. Perhaps, it would be easier to understand if the authors state that “neighborhood disadvantage is negatively associated with life expectancy”.

Reply: We have resolved the reviewer’s concern. By talking in terms of life expectancy instead of mortality.

I change the order to bring the noun closer to the verb.   But it is not just life expectancy and neighborhood disadvantage.  We also want to determine the relationship between poor air quality, economic distress and violent crime rates.

Page 2, line 91: It should be “negative effects”, not “a negative effects”.

Reply: We revised it accordingly.

Page 3, line 105: It should be “obtain”, not “obtained”.

Reply: We revised it accordingly.

Page 4, line 168: Perhaps, it should be “….or the end of the observation period in which they are alive”.

Reply: We revised it accordingly.

Page 5, line 191: It should be “…time they had the policy or had died”.

Reply: We revised it accordingly.

Page 5, line 219: It should be “…display the mean values of the area level variables in table 2…”.

Reply: We revised it accordingly.

Page 5, line 229: It should be “….younger than stayers…”

Reply: We revised it accordingly.

Page 8, line 240: Similar to my previous comments on the negative impact of neighborhood disadvantage on mortality.

Reply: Please see our reply above.

Page 14, Table 7: In Table, the authors show that in both unweighted and weighted models, the hazard of dying is largest for those who make >$75,000 annually. However, in Tables 5A and 5B, the results suggest that the higher the income a person is, the lower the hazard of dying for that person is. This is consistent with the results suggested in Table 7.

Could the authors provide some explanations to reconcile this?

Reply: We have added some explanations in the table header.

Page 16: The authors should have a separate section for conflicts of interest rather than having it in the same section with acknowledgement.

Reply: We revised it accordingly.